# Invasive Aspergillosis with Intracranial Extension Initially Misdiagnosed as a Granulomatous Disease: A Case Report

**DOI:** 10.3390/jof11070468

**Published:** 2025-06-20

**Authors:** Kouichi Asahi

**Affiliations:** 1Department of General Medicine and Radiology, Dokkyo Medical University Saitama Medical Center, Koshigaya 343-8555, Japan; kouichi81555@yahoo.co.jp; 2Kohokuekimae Ohisama Clinic, Internal Medicine and Pediatrics, Adachi 123-0872, Japan

**Keywords:** invasive aspergillosis, intracranial extension, fungal sinusitis, misdiagnosis, granulomatous inflammation, central nervous system infection, voriconazole, neuroimaging

## Abstract

**Background:** Invasive aspergillosis with orbital apex and intracranial involvement is rare and often misdiagnosed due to nonspecific imaging findings. Misinterpretation may lead to inappropriate therapies, such as corticosteroids, which can exacerbate fungal infections. **Case Presentation**: A 50-year-old immunocompetent woman with diabetes mellitus presented with right ptosis and systemic malaise. Magnetic resonance imaging (MRI) performed three months prior had shown a subtle low-signal lesion in the right orbital apex. The lesion was small and thought to represent a granulomatous process, with minimal systemic inflammation and only mild surrounding changes on imaging. Biopsy was considered too invasive at that stage, and the patient was placed under observation. Over time, her condition progressed, and repeat imaging revealed intracranial extension, including involvement of the cavernous sinus and frontal lobe. Differential diagnoses included granulomatous diseases such as sarcoidosis or tuberculosis, prompting empirical anti-tuberculosis treatment. However, the patient’s condition worsened, and biopsy of the sphenoid sinus revealed septated fungal hyphae consistent with Aspergillus species on Grocott staining. Voriconazole therapy was initiated, resulting in significant clinical and radiological improvement. **Discussion:** This case highlights the diagnostic challenge of identifying orbital apex aspergillosis with early MRI changes and demonstrates the risk of misdiagnosis as granulomatous disease. Differentiating fungal infections from other inflammatory etiologies based on subtle imaging features is critical, especially when considering immunosuppressive therapy. **Conclusion:** Clinicians should maintain a high index of suspicion for fungal infections in patients with progressive orbital apex lesions, even in the absence of classic immunosuppression. Early imaging review and biopsy are essential to prevent misdiagnosis and inappropriate treatment.

## 1. Introduction

Invasive aspergillosis (IA) is a life-threatening opportunistic fungal infection that predominantly affects immunocompromised individuals. However, emerging evidence indicates that IA can also develop in patients with only mild or localized immune dysfunction, such as those with diabetes mellitus, chronic lung disease, or advanced age [1,2]. Although the lungs are the most common site of involvement, extrapulmonary manifestations—including those affecting the paranasal sinuses, orbital apex, and intracranial compartments—are increasingly recognized, albeit rare [3].

The orbital apex is a complex anatomical region containing the optic canal and superior orbital fissure, where cranial nerves and vascular structures converge. Infections in this area may lead to orbital apex syndrome, characterized by ophthalmoplegia, ptosis, and visual disturbances. Fungal infections in this region, especially those caused by Aspergillus species, pose a diagnostic challenge due to their insidious onset, subtle radiological findings, and nonspecific clinical manifestations.

In immunocompetent patients, the disease may progress more slowly and mimic non-infectious inflammatory disorders such as sarcoidosis, granulomatosis with polyangiitis, or tuberculous granulomas [4,5]. Imaging findings such as low signal intensity on T2-weighted MRI or bone erosion on CT may offer early diagnostic clues, but these are not pathognomonic. Moreover, the decision to perform a biopsy in this anatomically delicate region is often deferred, especially when inflammatory markers are unremarkable, resulting in delayed diagnosis.

The risk of administering corticosteroids or other immunosuppressive treatments before confirming the infectious etiology can lead to rapid deterioration, particularly if fungal pathogens are present. Prompt identification and initiation of antifungal therapy are essential, yet often delayed due to the deceptive clinical picture. Recent reviews have emphasized the importance of including fungal infections in the differential diagnosis of orbital apex lesions, even in patients without overt immunosuppression [6,7].

This report presents a rare case of invasive orbital apex aspergillosis with contiguous intracranial extension in an immunocompetent patient. The case underscores the importance of maintaining a high index of suspicion for fungal infections even in patients without classic risk factors, and highlights the critical role of early biopsy and targeted antifungal therapy in optimizing patient outcomes.

## 2. Case Presentation

The patient was a 50-year-old Japanese woman with well-controlled type 2 diabetes mellitus (HbA1c 5.8%) and no history of immunosuppressive therapy. She was a non-smoker and worked as an office administrator. Three months prior to presentation, she began experiencing mild retro-orbital pain, fatigue, and intermittent low-grade fever. At that time, her symptoms were attributed to nonspecific viral illness. However, she gradually developed right-sided ptosis and complained of diplopia on lateral gaze, prompting ophthalmologic referral.

Initial neurological examination revealed partial right oculomotor nerve palsy with preserved pupillary light reflex and no afferent pupillary defect. Visual acuity was 1.0 (20/20) bilaterally. Fundoscopic examination was unremarkable, with no signs of papilledema or retinal changes. Laboratory findings were within normal limits, including white blood cell count (6800/μL), C-reactive protein (0.2 mg/dL), and erythrocyte sedimentation rate (12 mm/h). Autoimmune and infectious panels, including ANA, ACE, quantiferon, and HIV, were negative.

Orbital and brain MRI demonstrated a faint low-signal lesion in the right orbital apex on T2-weighted images, with mild gadolinium enhancement on fat-suppressed T1-weighted images. No surrounding edema or sinus involvement was noted. The lesion was interpreted by the neuroradiologist as likely granulomatous in nature, with a differential that included sarcoidosis and idiopathic orbital inflammation. Given the lesion’s deep location and absence of systemic findings, biopsy was deferred, and clinical observation was initiated (Figure 1A).

Over the following weeks, her symptoms progressed. She reported increasing difficulty in eye movement, blurred vision in the right eye, and worsening retro-orbital pain. Visual acuity in the right eye decreased to 0.3 (20/60), while the left eye remained unaffected. New findings included proptosis and chemosis without conjunctival injection. Repeat MRI revealed lesion progression into the right cavernous sinus and frontal lobe, with peripheral ring enhancement and central diffusion restriction, suggesting abscess formation. Edema in the right frontal lobe was also observed (Figure 1B, Figure 2).

CT of the sinuses demonstrated opacification of the right ethmoid and sphenoid sinuses, with bony erosion of the medial orbital wall and sinus roof (Figure 3). These findings raised suspicion for an invasive fungal process.

Empirical treatment with anti-tuberculosis therapy (isoniazid 300 mg/day, rifampicin 450 mg/day, ethambutol 750 mg/day, and pyrazinamide 1000 mg/day) and broad-spectrum antibiotics (ceftriaxone 2 g/day) was initiated. However, the patient experienced gastrointestinal side effects including nausea and anorexia, and her symptoms continued to worsen.

Transnasal endoscopic biopsy of the sphenoid sinus was performed. Histopathology revealed necrotizing granulomatous inflammation. Grocott methenamine silver staining showed septated fungal hyphae with acute-angle branching, confirming the diagnosis of Aspergillus species (Figure 4).

Antifungal therapy was initiated with intravenous voriconazole (6 mg/kg every 12 h for the first two doses, followed by 4 mg/kg every 12 h). The patient showed gradual improvement in ocular symptoms and resolution of fever. Follow-up MRI after one and three months showed a marked reduction in lesion size. The voriconazole course was continued for 12 weeks, during which liver function remained within acceptable limits, and no severe side effects were reported. At six-month follow-up, the patient remained symptom-free with normal visual acuity and full ocular motility.

## 3. Discussion

Invasive fungal infections (IFIs) of the orbital apex with contiguous intracranial involvement are rare and frequently misdiagnosed, particularly in non-neutropenic hosts. While traditionally considered diseases of the severely immunocompromised, increasing case reports—such as those by Sharma et al. and DeShazo et al.—demonstrate that even patients with localized immunosuppressive risk factors (e.g., diabetes mellitus) are susceptible to aggressive disease progression [1,2,3]. In our case, the patient was immunocompetent aside from well-controlled diabetes and presented with imaging findings that were initially subtle and non-specific.

The differential diagnosis for orbital apex lesions is broad, including granulomatous diseases (e.g., sarcoidosis and tuberculosis), neoplasms (e.g., lymphoma and meningioma), and idiopathic orbital inflammatory syndrome (IOIS). These entities often overlap radiologically, complicating the diagnosis. Sarcoidosis tends to be bilateral and systemic with elevated ACE levels, whereas tuberculosis often presents with basal meningeal enhancement and positive interferon-gamma release assays (IGRAs). In our case, negative IGRA and the absence of systemic granulomatous signs made tuberculosis less likely, yet empiric anti-TB treatment was initiated due to the ambiguous clinical picture. MRI plays a central role in early detection, but fungal lesions may lack definitive characteristics. Typical T2 hypointensity due to paramagnetic elements like iron and manganese can be absent early in the disease. Gadolinium enhancement may appear mild or irregular, as observed in our case. CT, particularly in bone windows, can reveal erosive changes in the sinus walls or orbital apex, suggestive of invasive fungal sinusitis. Diffusion-weighted imaging (DWI) is particularly valuable for detecting fungal abscesses through diffusion restriction patterns, while PET/CT may help distinguish neoplastic from infectious lesions, albeit with limited specificity and accessibility [4,5,6,7].

Histopathological confirmation remains essential. In our patient, transnasal sphenoid sinus biopsy was critical in establishing the diagnosis. Grocott and PAS staining revealed acute-angle branching hyphae typical of Aspergillus, allowing prompt initiation of antifungal therapy. While fungal cultures or molecular diagnostics such as ITS sequencing can provide species identification and susceptibility, clinical urgency often necessitates empiric therapy [8].

Voriconazole remains the treatment of choice due to its superior efficacy and CNS penetration. Therapeutic drug monitoring (TDM) is recommended to maintain therapeutic levels and minimize toxicity such as visual disturbances or hepatotoxicity. Although TDM was not performed in our case, the patient tolerated the drug well and exhibited significant clinical improvement without adverse effects [2]. This reinforces the importance of early treatment, particularly when diagnosis is delayed.

This case also highlights the importance of multidisciplinary collaboration. Involvement of infectious diseases, ophthalmology, otolaryngology, and radiology facilitated appropriate decision-making and expedited biopsy and treatment. As emphasized in recent guidelines, IFIs in non-neutropenic patients remain a diagnostic challenge and require high clinical suspicion, early recognition, and prompt therapy to optimize outcomes [9].

## 4. Conclusions

Clinicians should maintain a high index of suspicion for invasive fungal infections in patients presenting with progressive orbital apex lesions, particularly when imaging is atypical or symptoms persist despite empirical treatment. Diagnostic consideration should be heightened in the following scenarios:

Gradual worsening of cranial neuropathies (e.g., ptosis and ophthalmoplegia);

Poor response to corticosteroids or anti-tuberculosis therapy;

T2-hypointense orbital lesions with subtle enhancement on MRI;

Evidence of bony erosion or sinus involvement on CT;

Elevated serum fungal markers (β-D-glucan and galactomannan).

This case underscores the importance of early biopsy and multidisciplinary collaboration in achieving accurate diagnosis and effective treatment. It also highlights that invasive aspergillosis can develop even in immunocompetent patients, especially those with diabetes mellitus. Prompt recognition and targeted antifungal therapy are essential to prevent irreversible complications, such as vision loss or intracranial dissemination. Our report contributes to the growing awareness of this diagnostic pitfall and emphasizes a systematic approach for earlier identification and intervention.

## Figures and Tables

**Figure 1 jof-11-00468-f001:**
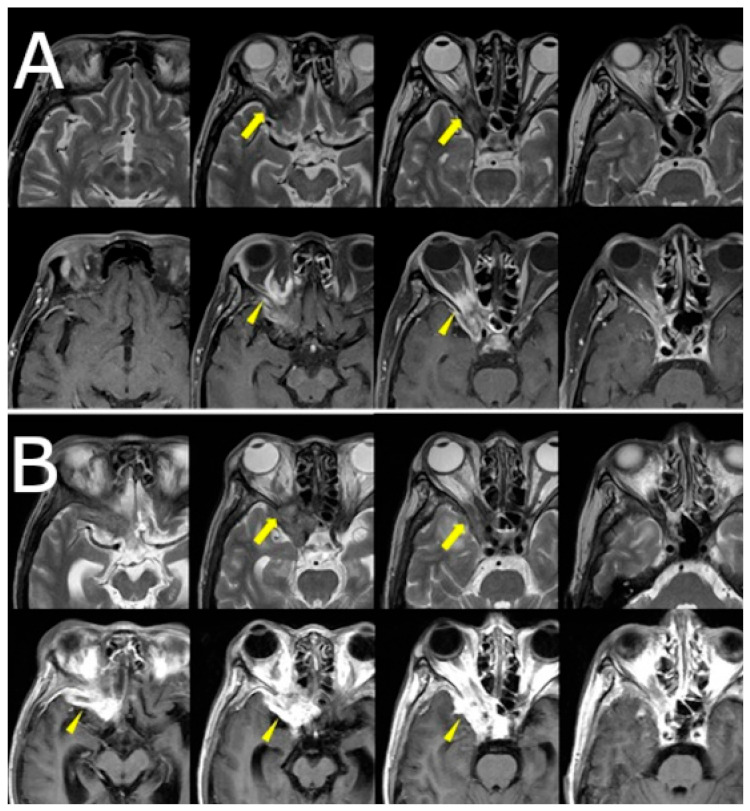
(**A**) Axial T2-weighted images (top row) and gadolinium-enhanced fat-suppressed T1-weighted images (bottom row) at the initial visit show a subtle low-signal lesion in the right orbital apex (yellow arrows). (**B**) Follow-up axial T2-weighted images (top row) and gadolinium-enhanced non-fat-suppressed T1-weighted images (bottom row) demonstrate progression of the lesion into the right orbital apex with increased enhancement and slight invasion of the adjacent cavernous sinus (yellow arrows).

**Figure 2 jof-11-00468-f002:**
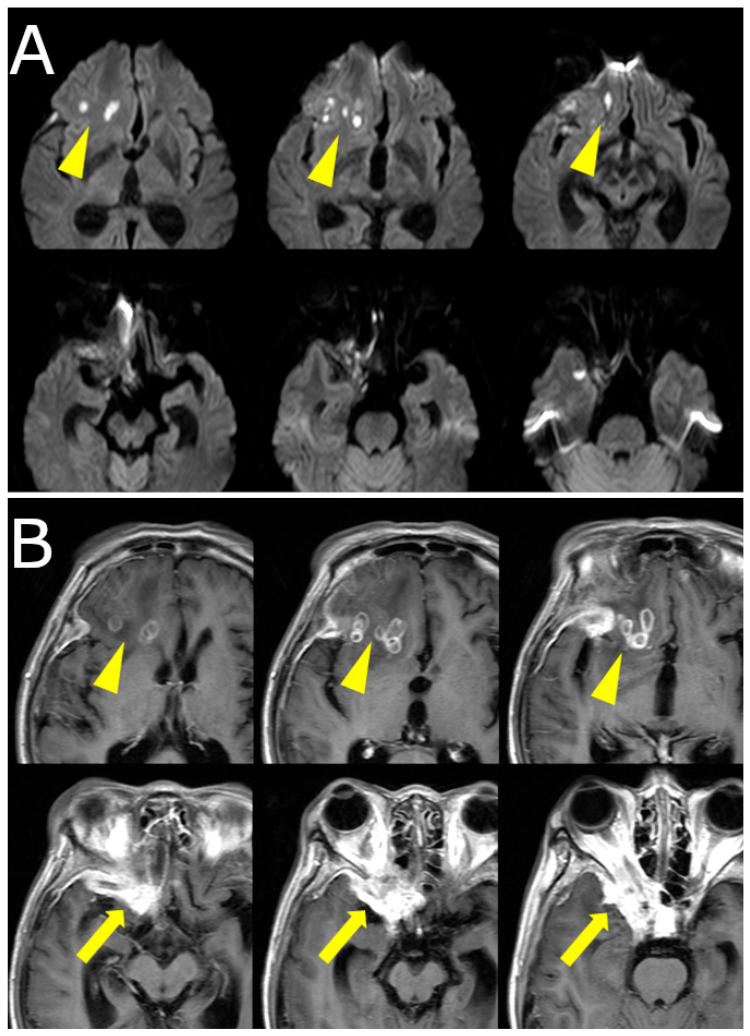
(**A**) Diffusion-weighted MRI (DWI) showing restricted diffusion in the right frontal lobe and orbital apex, indicative of abscess formation (yellow arrowheads). (**B**) Contrast-enhanced T1-weighted MRI demonstrating ring-enhancing lesions in the frontal lobe (arrowheads) and intense enhancement in the orbital apex and cavernous sinus (yellow arrows), consistent with intracranial extension.

**Figure 3 jof-11-00468-f003:**
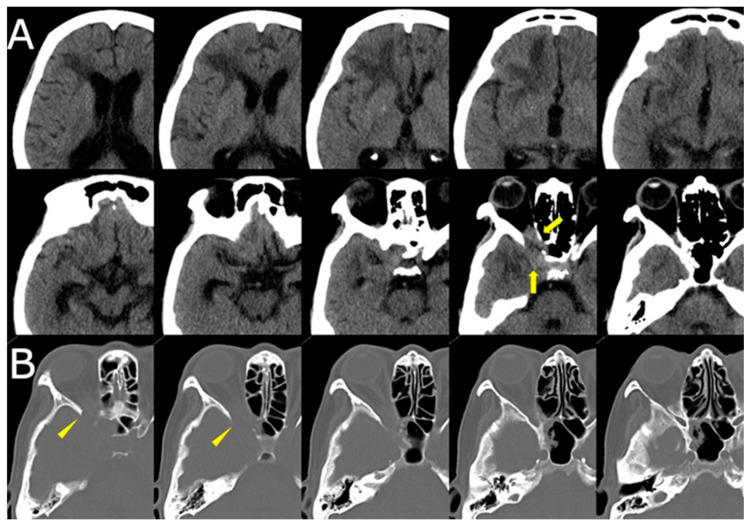
(**A**) Axial CT (brain window) demonstrating hypodense changes in the right frontal lobe with subtle soft tissue density in the right orbital apex and ethmoid sinus (yellow arrows). (**B**) CT (bone window) revealing bony erosion of the medial orbital wall and ethmoid sinus, suggesting invasive fungal involvement (yellow arrowheads).

**Figure 4 jof-11-00468-f004:**
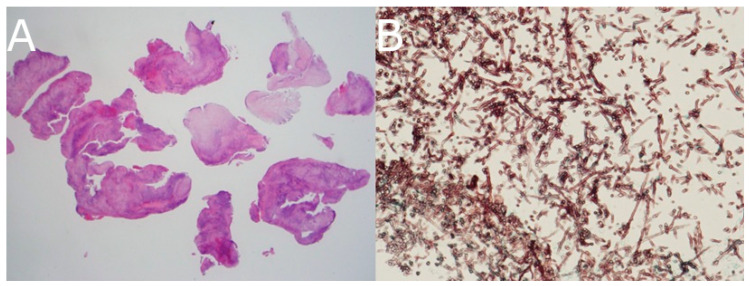
(**A**) Hematoxylin and eosin (H&E) staining of the sphenoid sinus biopsy specimen demonstrating necrotizing granulomatous inflammation (original magnification ×12.5). (**B**) Grocott methenamine silver stain (original magnification ×400) revealing numerous slender, septated fungal hyphae with acute-angle branching, characteristic of Aspergillus species.

## Data Availability

The raw data supporting the conclusions of this article will be made available by the author on request.

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
