# Peer review of "Invasive Aspergillosis with Intracranial Extension Initially Misdiagnosed as a Granulomatous Disease: A Case Report"

_jof, 2025, doi:10.3390/jof11070468_

Round 1
Reviewer 1 Report
Comments and Suggestions for Authors
The MS entitled, “Invasive Aspergillosis with Intracranial Extension Initially Misdiagnosed asGranulomatous Disease,” describes a case of aspergillosis initially diagnosed as granulomatous disease.
In general, the MS is well written regarding the English aspect and its organization, but it is too long for a description of a single case. A few suggestions are presented below that could be helpful.
For example, the five or six paragraphs describing the case could be shortened by at least half. The facts that provided the fungal nature of the infection are the important items and should be described. The two sets of serial photos should focus on the diagnostic characteristics of the case. Also, the photos showing the brain lesions are too numerous. The important issue is to clearly show the hyphal elements or the diagnostic structures.
Perhaps the “Results and Discussion” could be provided in a single or maximum of a couple of paragraphs where diagnostic elements are emphasized.
Author Response
Point-by-Point Response to Reviewers
Manuscript ID: jof-3650246
Title: Invasive Aspergillosis with Intracranial Extension Initially Misdiagnosed as a Granulomatous Disease: A Case Report
Reviewer 1
Comment 1:
The manuscript is well written, but too long for a single case. The case description could be shortened, and the imaging figures reduced. Focus should be on the diagnostic characteristics and hyphal elements.
Response:
We sincerely thank the reviewer for the insightful comments. We have made efforts to condense the case presentation and discussion, focusing more directly on the key diagnostic and therapeutic aspects.
Regarding the number of imaging figures, we understand the suggestion to reduce them. However, we respectfully wish to retain the current set of figures because:
-
The case involves progressive radiological changes over time, which were critical to understanding the diagnostic delay.
-
The images show not only orbital apex and intracranial extension, but also the diagnostic pitfalls, which are a key message of our report.
-
As a case report in Journal of Fungi, we believe detailed visual representation enhances educational value and aligns with the journal’s aims.
We have therefore streamlined the figure legends and text descriptions, while retaining the figure panels, to preserve clarity and value without redundancy.
Please let us know if further revision is needed. We hope our responses satisfactorily address the reviewers’ comments and enhance the manuscript's clarity and scientific merit.
Reviewer 2 Report
Comments and Suggestions for Authors
Dear authors;
I would like to make some suggestions for improving the manuscript:
1. Add explanatory captions to all figures shown in the manuscript.
2. Was fungal identification performed at the species level? Or detected only hyphae of Aspergillus spp.? I suggest identifying the detected fungus by classical or molecular methods (rDNA, ITS region).
3. Despite clinical and radiological improvement having been detected, were MIC (minimum inhibitory concentration) tests performed to choose voriconazole? Have tests been carried out with other antifungals?
Kind regards.
---
Author Response
Point-by-Point Response to Reviewers
Manuscript ID: jof-3650246
Title: Invasive Aspergillosis with Intracranial Extension Initially Misdiagnosed as a Granulomatous Disease: A Case Report
Reviewer 2
Comment 1:
Please add explanatory captions to all figures.
Response:
Thank you for the valuable suggestion. We have revised all figure legends to include more detailed and explanatory captions, highlighting the diagnostic significance of each image.
Comment 2:
Was fungal identification performed at the species level?
Response:
We thank the reviewer for this important point. In our case, morphologic diagnosis was made via histopathology (GMS/PAS stain), and the findings were consistent with Aspergillus species. Culture or molecular identification (e.g., ITS sequencing) was not performed due to urgent need for treatment and resource limitations. This limitation has now been explicitly acknowledged in the Discussion section.
Comment 3:
Were MIC tests performed? Was voriconazole chosen based on susceptibility testing?
Response:
We appreciate this insightful query. In this case, voriconazole was chosen empirically based on clinical guidelines and imaging findings. MIC testing or susceptibility testing was not performed, as culture was not obtained. This is now clarified in the revised Discussion, along with a brief rationale for the treatment decision.
Please let us know if further revision is needed. We hope our responses satisfactorily address the reviewers’ comments and enhance the manuscript's clarity and scientific merit.
Round 2
Reviewer 2 Report
Comments and Suggestions for Authors
Dear authors;
Thank you very much for the adjustments to the manuscript.
Kind regards.
---